# Fatty Acid Supplementation Affects Skin Wound Healing in a Rat Model

**DOI:** 10.3390/nu14112245

**Published:** 2022-05-27

**Authors:** Alica Hokynková, Marie Nováková, Petr Babula, Miroslava Sedláčková, Hana Paulová, Miroslava Hlaváčová, Daniela Charwátová, Tibor Stračina

**Affiliations:** 1Department of Burns and Plastic Surgery, Faculty Hospital Brno and Faculty of Medicine, Masaryk University, Jihlavská 20, 625 00 Brno, Czech Republic; alicah@post.cz; 2Department of Physiology, Faculty of Medicine, Masaryk University, Kamenice 5, 625 00 Brno, Czech Republic; majka@med.muni.cz (M.N.); babula@med.muni.cz (P.B.); d.charwatova@gmail.com (D.C.); 3Department of Histology and Embryology, Faculty of Medicine, Masaryk University, Kamenice 3, 625 00 Brno, Czech Republic; msedl@med.muni.cz; 4Department of Biochemistry, Faculty of Medicine, Masaryk University, Kamenice 5, 625 00 Brno, Czech Republic; hpaulova@med.muni.cz (H.P.); hlavacova@med.muni.cz (M.H.)

**Keywords:** polyunsaturated fatty acids, wound, healing, oxidative stress, 4-hydroxy-2-nonenal

## Abstract

Polyunsaturated fatty acids (PUFA) play an important role in reparative processes. The ratio of PUFAs n-3 to n-6 may affect wound healing. The study aimed to evaluate the effect of dietary supplementation with n-3 and n-6 PUFA in two proportions on skin wounds in laboratory rats. Adult male Wistar rats received 20% fat emulsion with a ratio of 1.4:1 (group A) or 4.3:1 (group B) for n-3:n-6 PUFAs at a daily dose of 1 mL/kg. The control group received water under the same conditions. The animals were supplemented a week before and a week after the skin excision performed on the back. The level of wound closure, various parameters of oxidative stress, and plasma fatty acids composition were evaluated. Wound tissue samples were examined by electron microscopy. The administration of fat emulsions led to significant changes in plasma polyunsaturated fatty acid composition. The increased production of reactive nitrogen species, as well as more numerous newly formed blood vessels and a greater amount of highly organized collagen fibrils in both groups A and B may indicate more intensive healing of the skin wound in rats supplemented with polyunsaturated fatty acids in high n-3:n-6 ratio.

## 1. Introduction

Healing wounds of various origins represent a challenge in daily clinical practice and one of the biggest economic burdens for the health system, especially in hospitalized patients [1].

Wound healing represents a complex biological process on a cellular and subcellular level, with numerous intercellular interactions and activation of various chemokines, cytokines, and growth factors. Wound healing always proceeds in the following phases: inflammation, proliferation (granulation), and maturation (remodelling). For each of the healing phases, different cells and their products are typical and/or dominant, and their dominance overlaps [2].

The rate and quality of wound healing depends on numerous factors, including nutrition [3]. The effects of certain nutrients on wound healing have been studied, e.g., arginine [4], glutamine [5], vitamin C [6], and fatty acids (FA) [7]. From all the above mentioned, polyunsaturated fatty acids (PUFA) n-3 modulate the production of pro-inflammatory cytokines (IL-1, IL-6, TNF-alfa) by their end products (such as leukotrienes, prostaglandins, and thromboxane) and thus exert dominant anti-inflammatory effects [8]. In contrast, the effect of n-6 PUFA is rather pro-inflammatory [9]. The protective effects of n-3 PUFA on the cardiovascular system [10], atherogenesis [11], lipidaemia, immune systems, and certain neurological [12] and psychiatric disorders have been studied repeatedly. The role of n-9 PUFA should not be ignored, as these FA may affect the speed of wound closure [13].

In addition to human studies, animal models are used in wound healing studies. Local and systemic effects of the topical application of essential fatty acid oil on skin wounds were described in the rat model [14,15]. Recently, differences in tissue response during wound healing were reported in a rat model after supplementation with various dietary oils [16,17].

The ratio of PUFA n-3 to n-6 in diet leads to the dominance of a certain type of end-products of their metabolism (based on their mutual metabolic pathways) and as a result to pro-inflammatory or anti-inflammatory effects. The effects of PUFAs administration at different ratios and of various durations were reported recently in animal models [18,19]. The aim of the present study was to evaluate the effect of dietary supplementation with n-3 and n-6 PUFA in two different ratios on skin wound healing in laboratory rats.

## 2. Materials and Methods

### 2.1. Experimental Protocol

All experiments were carried out according to the recommendations of the European Community Guide for the Care and Use of Laboratory Animals and according to the experimental protocol (No. MSMT-35672/2016-2) approved by the Committee for Ensuring the Welfare of Laboratory Animals of Masaryk University and licensed by the Ministry of Education, Youth and Sports of the Czech Republic. The animals were housed at the Animal Breeding and Experimental Facility, Faculty of Medicine, Masaryk University in a temperature-, pressure- and humidity-controlled environment, with light cycle 12/12 (light/dark).

A total of 30 Wistar male rats (3 weeks old) were included in the study, with an average body mass of 160.4 ± 12.6 g. Figure 1 shows the scheme of the experimental protocol. The animals were handled for seven days (handling phase) to adapt to application with orogastric tube. Then they were randomly divided into three groups (A, B, and C; n = 10) and housed individually in cages with an enriched environment. All animals had *ad libitum* access to water and a standard diet throughout the experiment. Each rat was daily weighed using KERN 440-43N (KERN & Sohn GmbH, Balingen, Germany). Body mass increments were plotted and statistically evaluated. All manipulations and applications were performed during morning hours.

Seven days before surgery (pre-operative phase), animals in groups A and B received 20% vegetable-derived lipid emulsions A and B (Biomedica Praha, Praha, Czech Republic) by orogastric tube once a day (0.2 g/kg of actual body weight). The content of the main FA in the lipid emulsions A and B is summarized in Table 1. The ratio of n-3:n-6 PUFA was 1.4:1 in the emulsion A and 4.3:1 in the emulsion B, respectively. Animals in the control group (C) received *water for injection* in corresponding volume (1 mL/kg of actual body weight).

On day 14, the animals were deeply anaesthetized with ketamine (100 mg/kg, i.p.) and xylazine (10 mg/kg, i.p.). The interscapular area was shaved and a round full-thickness skin excision (2 cm in diameter) was performed. The wound was left without dressing to heal *by second intention*. During the following 7 days (postoperative phase), daily administration of lipid emulsion or water continued under the same conditions.

### 2.2. Analysis of Wound Closure Using Digital Planimetry

The daily area that is yet to heal was measured by a noncontact method using a tablet with specialised planimetric software (Electreasure, HC Electronics, Hradec Králové, Czech Republic). The first measurement was performed immediately after surgery and then the measurements were made daily. The wound snapshot together with the calibration ruler was taken with the built-in camera (Appendix A). Each wound snapshot was calibrated according to the ruler and stored. The wound perimeters were then outlined on the display with a stylus, and the area that had not yet healed (in cm^2^) was calculated. The level of wound closure was calculated as follows.
WCL = (WA_D14_ − WA_D21_)/WA_D14_ × 100 [%],(1)
where WCL—wound closure level; WA_D14_—wound area on day 14 (immediately after the surgery); WA_D21_—wound area in the day 21 (day of the termination).

### 2.3. Sample Collection

On day 21, the animals were deeply anaesthetized, and blood samples were collected by direct intracardial puncture with heparin or EDTA. The animals were sacrificed and the tissue samples from the wound border were promptly excised (Figure 2b).

### 2.4. Hematological Analyses

Basic hematological parameters—namely red blood cell count, hematocrit, hemoglobin concentration, red blood cell distribution width, platelets count, and white blood cell count were evaluated from EDTA blood samples. The analysis was performed immediately after sample collection using the Mythic 18 blood analyzer (Orphée SA, Plan-les-Ouates, Switzerland). Each sample was measured twice and the average was used for statistical evaluation.

### 2.5. Biochemical Analyses

Blood plasma was separated from the heparinised blood sample and stored for further processing at −80 °C. Plasma fatty acid composition was evaluated by gas chromatography as previously described [20,21]. Various plasma oxidative stress parameters were evaluated, including determination of 4-hydroxy-2-nonenal (4-HNE), hydrogen peroxide, nitrite/nitrate ratio, total antioxidant capacity, total reactive nitrogen species, and total reactive oxygen species. The measurement of 4-HNE levels was performed by HPLC as previously described [22] by applying the reaction with 2,4-dinitrophenylhydrazine (DNPH) as derivatization reagent. The fluorometric hydrogen peroxide assay kit (MAK165, Sigma Aldrich, St. Louis, MO, USA) used to determine hydrogen peroxide. To determine the nitrite/nitrate ration, a colorimetric nitrite/nitrate assay kit (23479, Sigma Aldrich, USA) was used. The colorimetric total antioxidant capacity assay kit (MAK187, Sigma Aldrich, USA) was used to evaluate the total antioxidant capacity of blood plasma. All kits were used according to the manufacturer’s instructions. The total level of reactive nitrogen species (RNS) was measured using a fluorescence probe 2,3-diaminonaphthalene (DAN, λex = 365/λem = 415 nm, D2757, Sigma Aldrich, USA) according to Rao et al. [23]. The total level of reactive oxygen species (ROS) was measured using a fluorescent probe 2′,7′-dichlorodihydrofluorescein (H_2_DCF, λex = 495/λem = 527 nm, D6665, Sigma Aldrich, USA), according to Orozco-Ibarra et al. [24]. The Cytation 3 reader (BioTek Instruments, Winooski, VT, USA) was used for all colorimetric and fluorometric measurements.

### 2.6. Tissue Sample Analyses

#### 2.6.1. Fluorescence Microscopy

The presence of ROS in wound samples (approximately 3 × 3 × 3 mm) was visualized using fixable CellROX™ Green Reagent (λex = 485/λem = 520 nm, C10444, Thermo Fisher Scientific, Waltham, MA, USA). The RNS were visualized using 2,3-DAN (10 μM in 0.625 M HCl) as a fluorescent probe. Cell nuclei were counterstained with propidium iodide (1 μM solution in 0.9% NaCl, P4170, Sigma Aldrich, USA). Mito Tracker Red FM (750 nM, λex = 581/λem = 644 nm, M22425, Thermo Fisher Scientific, USA) was used to stain mitochondria. The nuclei were counterstained with 4′,6-diamidine-2′-phenylindole dihydrochloride (500 nM, 000000010236276001, Sigma Aldrich, USA). Acridine orange/ethidium bromide fluorescence staining (318337 and E7637, respectively; both chemicals Sigma Aldrich, USA) was used to evaluate cell morphology and apoptosis, respectively. The protocol according to Jimenez et al. was used [25]. The BODIPY™ 581/591 C11 lipid peroxidation sensor (1 μM, D3861, Thermo Fisher Scientific, USA) was used to determine the rate of lipid peroxidation in wound samples. Depending on the fluorescent probe used, the samples were directly observed or fixed in the fixation (R37602, Image-iT™ Fixation/Permeabilization Kit, Thermo Fisher Scientific, USA). An epifluorescence microscope Nikon Eclipse Ti-S/L100 (Nikon, Tokyo, Japan) and appropriate excitation/emission wavelengths were used for all observations. The NIS-elements software (Nikon, Japan) was used to process images and analyze the resultant pictures.

#### 2.6.2. Electron Microscopy

Small skin blocks (1–2 mm^3^) were fixed in 300 mM glutaraldehyde (Sigma Aldrich, USA) dissolved in 100 mM cacodylate buffer for 2 h at room temperature, washed and postfixed with 40 mM osmium tetroxide (Polysciences, Warrington, PA, USA) in the same buffer for 1 h at room temperature. After being rinsed in buffer and dehydration in ethanol, the samples were embedded in araldite resin (Durcupan ACM, Sigma Aldrich, USA). Precise block orientation was made to obtain a perpendicular section of the healing skin defect. Ultrathin sections (60 nm thick) were cut using a Leica EM UC6 ultramicrotome and stained with uranyl acetate and lead citrate. The sections were examined under a FEI Morgagni 268D transmission electron microscope (FEI Company, Hillsboro, OR, USA) at 70 kV.

### 2.7. Statistical Analyses

Statistical analyses were performed using GraphPad Prism 5 (GraphPad Software, San Diego, CA, USA). Data distribution was tested by the D’Agostino-Pearson normality test. Because of the non-Gaussian data distribution, nonparametric statistical analysis was employed. Between-group (group A vs. group B vs. group C) comparisons were evaluated using the Kruskal–Wallis’s test (nonparametric analysis of variance), followed by multiple comparison (Dunn’s test). Paired measurements were compared by the Wilcoxon matched pairs test. The results are presented as median (lower quartile–upper quartile) or as mean ± S.E.M., *p*-values < 0.05 were considered statistically significant. 

## 3. Results

### 3.1. Wound Closure Level

Figure 2a shows the mean level of wound closure in each group from the day of the surgery to the end of the experiment. In group A, the wound area was significantly enlarged on days 15 and 16 as compared to day 14. From day 17, the wounds started to retract. The same trend was observed in group B. In group C, a continual trend of wound closure was observed. At the end of the experiment, the highest level of wound closure was found in group C (57.94 ± 4.85%). In group B, the wound area decreased by 46.26 ± 5.25%. The lowest level of wound closure was detected in group A (35.35 ± 5.72%), significantly lower compared with group C. No significant differences in wound closure level were detected between groups B and C as well as groups A and B. Representative pictures of wounds are presented in Appendix A.

### 3.2. Blood and Plasma Parameters

The results of the hematological analyses are summarized in Appendix A. No significant differences were found in either of the hematological parameters.

The two-week administration of the lipid emulsion led to alterations in FAs occurrence in both plasma phospholipids and plasma triacylglycerols. A total of 11 PUFAs were analysed, namely 4 acids from the n-3 group (α-linolenic acid (ALA), eicosapentaenoic acid (EPA), docosapentaenoic acid (DPA) and docosahexaenoic acid (DHA)) and 7 acids from the n-6 group (linoleic acid (LA), γ-linolenic acid, dihomolinoleic acid, dihomo-γ-linolenic acid, arachidonic acid, adrenic acid, and osbond acid). Full results of the analyses are available in the Appendix A. The total amounts of PUFAs n-3 and n-6 were not significantly changed, although total amount of n-3 PUFA in triacylglycerols was close to significance (*p* = 0.061). However, significant changes in two (out of seven) n-6 PUFAs and one (out of four) n-3 PUFAs in phospholipids were observed (Table 2). Similar results were obtained in triacylglycerols (Table 3).

No significant differences in plasmatic levels of 4-HNE were found (10.0 ± 6.9 vs. 6.1 ± 3.4 vs. 5.6 ± 3.7 nmol/L in groups C, A, and B, respectively).

Colorimetric and fluorometric analyses of oxidative stress parameters in blood plasma revealed a significant decrease in the hydrogen peroxide and the nitrates/nitrites ratio in the A and B groups, with the lowest values in group B. Production of RNS and nitrites increased significantly in the A and B groups with the highest values in group B. ROS, total antioxidant capacity, and nitrate production were not significantly affected.

### 3.3. Wound Tissue Samples 

The examination of the parameters of oxidative stress by fluorescence microscopy (Figure 3 and Figure 4) corresponded to that obtained from blood plasma. In the control group, the ROS distribution showed accumulation in well-bounded areas as revealed by ROS visualization. In groups A and B, the number of stained nuclei was minimal compared to the control. Well-stained nuclei are evidence of minimal interference, and the presence of the artefacts in the case of the control group and diffuse fluorescence in groups A and B may indicate possible interference due to staining or release of the fluorescent product. The staining of RNS revealed their increased amount in tissues of groups A and B. Compared to the control and similar to ROS, RNS showed diffuse distribution in the A and B groups. These results are supported by visualization of lipid peroxidation products, which were well evident in distinct areas of the tissues of the controls (the ratio between the fluorescence of two wavelengths; a shift to green fluorescence indicates a greater degree of lipid peroxidation). Tissue samples from the control group demonstrated the presence of areas where apoptotic nuclei accumulated. On the other hand, in groups A and B, the apoptotic nuclei were evenly distributed within the tissue, which corresponds to the results obtained using mitochondrial potential staining (not shown). 

Electron microscopic analysis was performed in the region of the skin defect, where epithelization had begun, to a depth of approximately 500 µm. Morphology and relative representation of cellular elements (fibroblasts, macrophages, neutrophils, and lymphocytes), angiogenesis, and the structure of extracellular matrix (ECM) were evaluated.

Fibroblasts with a heavily developed rough endoplasmic reticulum and macrophages were the most numerous cell types found in all samples; neutrophils and lymphocytes were rarely found. The voluminous ECM contained fine collagen fibrils with varying degrees of organization. Compared to the control group, the number of fibroblasts was lower and the number of macrophages was higher in groups A and B. The newly formed blood vessels were more numerous in both groups (A and B), and the ECM contained a greater amount of collagen fibrils that began to organize into larger units (fibres). Thinner fibrils (40 nm) and thicker fibrils (80 nm) were observed (Figure 5). This may indicate a more intensive production of collagen in groups A and B (compared to the control group).

## 4. Discussion

Several authors have reported the progression and potentiation of the speed and quality of wound healing with respect to different content of PUFAs, namely n-3 and n-6 PUFAs in a diet [16,26,27]. However, the exact roles of these PUFAs and mainly their mutual relationship in wound healing have not been fully elucidated. 

The Western diet, typical for developed countries with a ratio of PUFAs of n-3 and n-6 as high as 1:20, leads to high incidence of cardiovascular diseases, obesity, atherosclerosis, and diabetes mellitus. The incidence of these diseases is lower in populations on a Mediterranean diet (fish products, olive oil, etc.), in which the ratio of PUFAs of n-3 and n-6 varies from 1:1 to 1:4. The protection of the cardiovascular system by n-3 PUFAs has already been known for several years [28].

In the present study, the effects of PUFA supplementation on skin wound healing in rats were studied. The experimental design is unique in several aspects. First, the lipid emulsion with two different ratios between the PUFA n-3 and n-6 was administered; in both emulsions the ratio was quite low, approaching the levels observed in a typical Mediterranean diet. Second, there is only a mild difference between both administered emulsions. Third, short-lasting administration of lipid emulsion was performed only a week before wound formation and a week before the experiment termination. In addition, plasma levels of FAs concentration and the specific lipid peroxidation marker 4-HNE were evaluated. To our knowledge, there is no experimental study taking into account both short duration of changed diet and the low ratio n3:n6 PUFAs at the same time. 

In the present study, two-week administration of lipid emulsion with an increased amount of n-3 PUFA leads to increased plasmatic level of n-3 PUFA in triacylglycerols, with borderline significance, which could be explained by the minor differences in the n-3:n-6 ratios in applied emulsion. Significant increases in EPA levels in plasma phospholipids and DHA values in plasma triacylglycerols were observed. 

In humans, it has been reported that four-week application of EPA/DHA in the dose of 1.6 and 1.2 g/day, respectively, leads to significant increases in plasma FA levels for both EPA and DHA [7], which was consistent with several previous studies [29]. On the other hand, a recent study by Mihalj et al. [26] reported a negligible effect of a three-week diet with a higher PUFA content on serum concentrations of n-3 and n-6 PUFA in humans.

The results of the present study correspond to most reports mentioned above. However, our results are barely comparable to the findings from previously reported animal studies. The reasons are different methodological approaches, e.g., long-term PUFA administration, different n-3:n-6 PUFA ratio with high content of n-6 PUFAs, etc. [16,18,19]. In addition, the level of FA was measured mainly in tissue, contrary to our study where plasma concentrations were evaluated.

In addition to plasma concentrations of FAs, reactive oxygen species and products of lipid peroxidation were evaluated. Reactive oxygen species (ROS), including the hydroxyl radical, are known to affect wound healing [30]. ROS was reported to affect the expression of some nuclear-encoded genes crucial for wound healing response and oxidative stress [31]. As expected, areas with ROS accumulation were detected both in the control and experimental groups. A secondary product of lipid peroxidation, 4-HNE, was also analysed as one of the oxidative stress parameters. The aforementioned 4-HNE comes from n-6 PUFAs (e.g., linoleic or arachidonic acid) [32] and is also known as a modulator of cell functions [33]. No significant differences in 4-HNE concentrations were found between the groups. However, the decreasing trend of 4-HNE in groups A and B suggests the positive effect of a higher intake of n-3 PUFAs on lipid peroxidation measured by this parameter. Another study confirmed that the higher intake of n-3 PUFA leads to a decrease in another marker of lipoperoxidation – malondialdehyde in mice tissue [27]. It should be noted that in the case of induced lipoperoxidation (e.g., doxorubicin administration [22]), the concentration of 4-HNE increases and 4-HNE is considered as a good marker reflecting the actual level of lipid peroxidation.

The observed changes in blood plasma were reflected in ultrastructural changes. Electron microscopy revealed a wide range of cell types in the wound healing area: elements that remove decomposed cells or infectious agents (macrophages, neutrophil granulocytes) or elements that participate in healing processes by producing an intercellular matrix (fibroblasts). No significant differences were found among groups A, B and C when representation of particular cell types was compared, most probably due to variability of the samples from the same group. Another possible explanation is the inhomogeneity of the healing area and also the fact that only a small portion of the tissue sample could be examined. In terms of the extracellular matrix, the structure of its fibrous component was studied. No elastic fibres were found in either sample, but all samples contained collagen in the form of microfibrils. In group B, difference in the quantity and quality of collagen was observed: collagen microfibrils were clearly organized into larger aggregates. In addition to very thin microfibrils (which were found in the samples from all groups), microfibrils of a bigger diameter were also observed. The finding probably reflects increased collagen production. This observation is in agreement with a previously published study reporting that n-3 PUFA stimulate fibroblasts to increase collagen production [34]. The significant increase in number of selected n-3 PUFAs in both experimental groups supports this conclusion. The question remains whether increased collagen production is beneficial or whether it might lead in the future to the formation of keloid scars.

The macroscopic picture of wound healing was studied by digital planimetry. Significant wound retraction was observed at the end of the experiment in all groups. However, in animals supplemented with emulsion with a lower content of n-3 PUFA (group A) the retraction was significantly smaller compared to the control group. A similar trend was observed in group B (supplemented by emulsion with a higher content of n-3 PUFAs). At this point, it is necessary to emphasize that the speed of wound closure is not the only or the most important parameter reflecting the healing process. Furthermore, the quality of the formed tissue must be taken into account. As discussed above, the ultramicroscopic picture of newly formed tissue in group B indicates that a higher content of n-3 PUFA stimulated formation of the collagen fibres of higher quantity and quality. The rather surprising widening of the wound area in groups A and B on days 15 and 16 is difficult to explain. However, it should be noted that this widening was to a smaller extent in group B, which was supplemented with a lipid emulsion with higher content of n-3 PUFA. 

As any other model, our experimental setup is burdened by certain limitations. In digital planimetry, the main drawback is the manual evaluation of the wound area. Although the same experimenter always performed the measurement, certain error must be considered since the wound diameter was small (in the range of millimetres) and the wound was not sharp-edged. This error may be minimized by employing automated wound area analysis [35]. In addition, it has to be considered that the results from animal models are not fully transferrable to human medicine. Most of the previously published studies that focused on the effects of PUFAs on wound healing were conducted in humans. However, the rat is a fully accepted model for such a type of study. Furthermore, the results of previously published studies are rather inhomogeneous, which may result from methodological differences, particularly from the different forms and durations of PUFA administration. If the results are compared, origin of lipids should also be considered. In the present study, the vegetable-derived lipid emulsions were administered contrary to most of the previously published studies, where animal-derived lipids were used.

## 5. Conclusions

It can be concluded that even a short-term administration of lipid emulsions containing higher n-3:n-6 PUFA ratio results in increased plasmatic level of n-3 PUFAs in triacylglycerols of borderline significance, significantly increase in EPA levels in plasma fosfolipids, and significantly increase in DHA plasma levels in triacylglycerols. The decreasing trend of 4-HNE supports this idea. The appearance of numerous newly formed blood vessels and a greater amount of highly organized collagen fibrils, revealed by electron microscopy of the skin defect, support the idea of higher quality wound healing. 

More studies are needed to uncover the particular mechanisms behind this beneficial effect of dietary supplementation with higher content of n-3 PUFAs.

## Figures and Tables

**Figure 1 nutrients-14-02245-f001:**
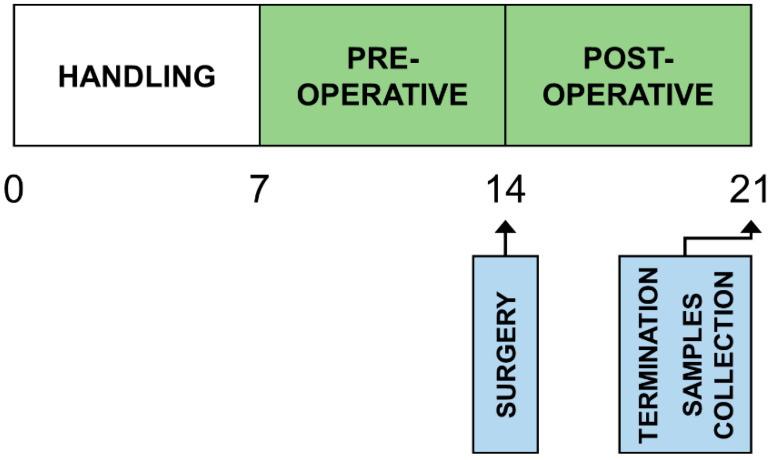
Scheme of the experimental protocol. The boxes represent the phases of the experiment; numbers represent the experimental days. The administration of the lipid emulsions is marked green.

**Figure 2 nutrients-14-02245-f002:**
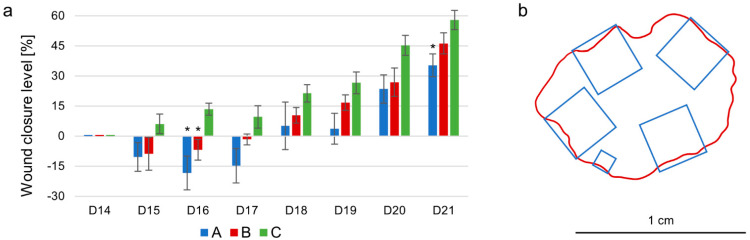
The wound closure level and tissue sampling. (**a**) The graph shows the wound closure level from day 14 (surgery) until day 21 (termination) in groups A (blue), B (red), and C (green), respectively. The columns represent the mean level of wound closure; the bars represent S.E.M.; * indicates statistical significance as compared to control group. (**b**) An example of tissue sample collection from the wound area. From each wound, 5 tissue samples were collected from the border zone; four of them for fluorescence microscopy (3 × 3 × 3 mm), one for electron microscopy (1 × 1 × 1 mm). The bar represents 1 cm.

**Figure 3 nutrients-14-02245-f003:**
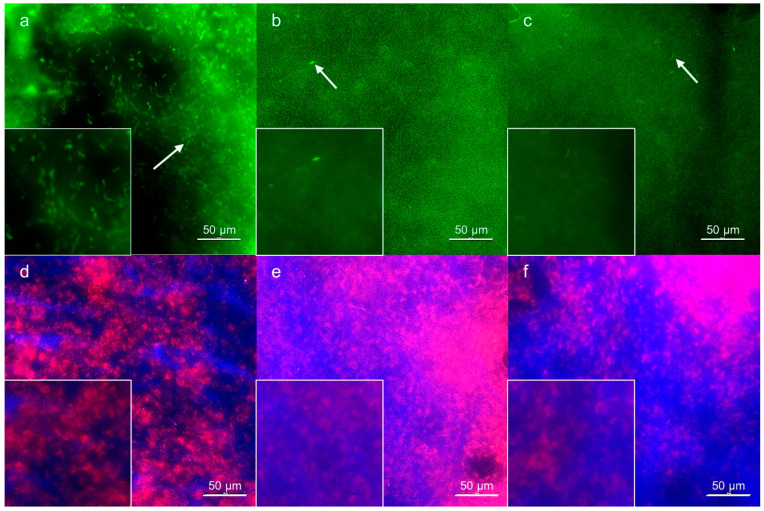
Representative fluorescence microscopy images showing the impact of PUFA administration on ROS/RNS in wound tissue samples. (**a**–**c**): CellROX™ Green Reagent staining; (**a**) control group, (**b**) group A, (**c**) group B. Green fluorescence is proportional to the amount of ROS accumulated in the tissue sample. Note the fluorescence of the nuclei (white arrows) in control group (**a**), group A (**b**), and B (**c**). (**d**,**e**): 2,3-diaminonaphthalene staining; (**d**) control group, (**e**) group A, (**f**) group B. The intensity of blue fluorescence corresponds to the amount of RNS in the tissue sample. The nuclei are counterstained with propidium iodide (red fluorescence). Note that in group B (**f**) the intensity of blue fluorescence overlays fluorescence of the stained nuclei. The insets represent magnified details. The bars represent 50 μm.

**Figure 4 nutrients-14-02245-f004:**
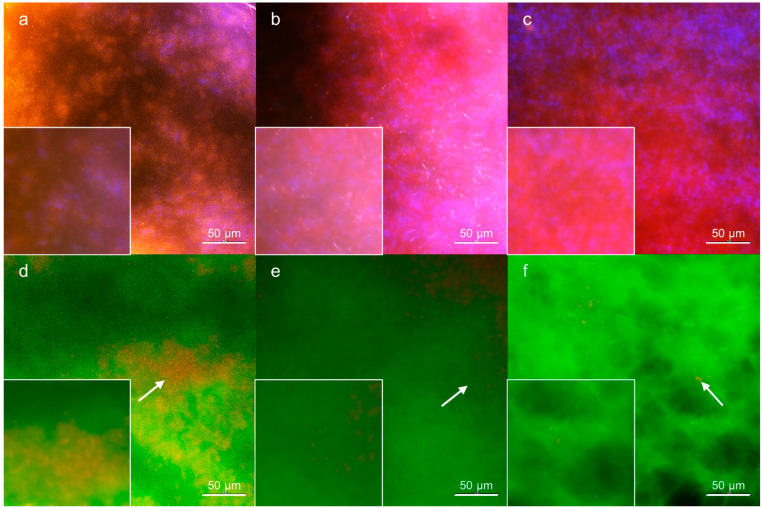
Representative fluorescence microscopy images showing the impact of PUFA administration on lipid peroxidation and apoptosis in wound samples. (**a**–**c**): BODIPY™ 581/591 C11 lipid peroxidation sensor; (**a**) control group, (**b**) group A, (**c**) group B. The rate of lipid peroxidation corresponds to the red/green fluorescence ratio. Nuclei were counterstained with DAPI (blue fluorescence). Staining indicates an increase in the lipid peroxidation in control group (**a**) compared to groups A (**b**) and B (**c**). (**d**,**e**): Acridine orange/ethidium bromide staining; (**d**) control group, (**e**) group A, and (**f**) group B. Staining visualizes apoptotic nuclei (red fluorescence). Note the clustering of apoptotic nuclei in the control group (**d**) and their relatively regular distribution in groups A (**e**) and B (**f**) (white arrows). The insets represent magnified details. The bars represent 50 μm.

**Figure 5 nutrients-14-02245-f005:**
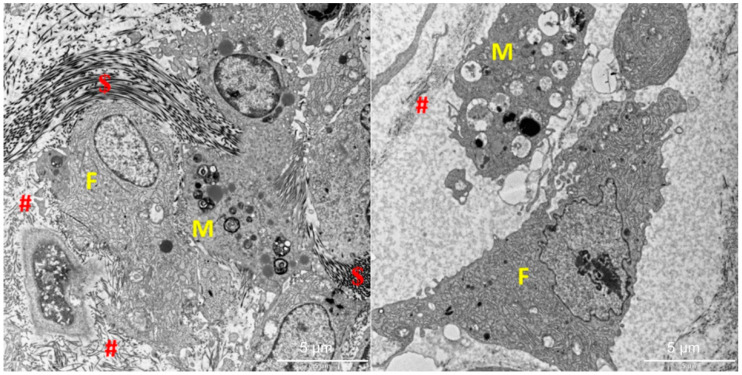
Representative electron microscopy images of healing skin defects in group A (**left**) and control group (**right**). Fibroblasts with heavily developed rough endoplasmic reticulum (F) and macrophages (M) were the most numerous cell types found in all samples. The voluminous extracellular matrix (ECM) contained collagen fibrils with a greater amount and higher degree of organization (fibers) in group A (**left**) and group B than in group C (#—ECM with collagen fibrils; $—collagen fiber). In groups A (**left**) and B, there were thinner fibrils (40 nm) and thicker fibrils (80 nm). This may indicate more intensive collagen production in groups A and B compared to the control group. The bars represent 5 μm.

**Table 1 nutrients-14-02245-t001:** Content of selected fatty acids in lipid emulsions. The numbers describe the content (in %) of the fatty acid in the experimental lipid emulsions A and B, respectively.

Lipid Emulsion	Emulsion A	Emulsion B
Palmitic acid	14.62	15.08
Stearic acid	3.07	2.73
Oleic acid	31.99	39.37
Linoleic acid (LA)	18.36	6.72
α-Linolenic acid (ALA)	15.32	16.74
Docosahexaenoic acid (DHA)	10.12	12.24

**Table 2 nutrients-14-02245-t002:** Significant changes in PUFAs in plasma phospholipids. The numbers represent the *p* values.

PUFA		Kruskal-Wallis ANOVA	Multiple Comparisons		
			C vs. A	C vs. B	A vs. B
Dihomo-γ-linolenic acid	20:3n-6	0.047 *	0.052	0.260	1.000
Osbond acid	22:5n-6	0.008 *	0.037 *	0.014 *	1.000
Eicosapentaenoic acid (EPA)	20:5n-3	<0.0001 ***	0.008 *	0.0001 ***	0.898

* *p* ≤ 0.05, *** *p* ≤ 0.0001.

**Table 3 nutrients-14-02245-t003:** Significant changes in PUFAs in plasma triacylglycerols. The numbers represent the *p* values.

PUFA		Kruskal-Wallis ANOVA	Multiple Comparisons		
			C vs. A	C vs. B	A vs. B
Linoleic acid (LA)	18:2n-6	0.0345 *	0.060	1.000	0.074
Osbond acid	22:5n-6	0.0267 *	0.176	0.029 *	1.000
Docosahexaenoic acid (DHA)	22:6n-3	0.048 *	1.000	0.047 *	0.368

* *p* ≤ 0.05.

## Data Availability

Not applicable.

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
