# Peer review of "Fatty Acid Supplementation Affects Skin Wound Healing in a Rat Model"

_nutrients, 2022, doi:10.3390/nu14112245_

Round 1

Reviewer 1 Report

The manuscript by Alica Hokynková et aims to describe the 

effect of fatty acid supplementation on skin wound healing in a rat model. Although the topic is interesting, there are however several issues that have to be addressed.

Introduction:

The introduction has to be re-written to include specific bibliography which supports the hypothesis the authors tested. The current format of the introduction is more suitable for a textbook article.

Results:

Several results are missing. I think it is necessary to see the rate of the healing process and not just pictures at two time points.

The titles of the subsections in the results should be indicative of the results and not of the technique used.

Where are the hematological results? The authors just mentioned that they found no differences.

Have the authors measured healing factors such as cytokines, TGFbeta, collagen accumulation etc?

Have the authors performed histological studies?

Discussion:

The discussion is too long and the authors do not discuss their findings in the context of the current literature.  

The whole manuscript has to be checked by an native English speaker.

Author Response

Dear Reviewer,

thank you very much for your careful review which substantially improved the quality of the revised text. Please see a point-to-point response to your comments in the attachment.

Reviewer 2 Report

This is a debated topic: wound healing and nutrition.

In this manuscript, the authors evaluate the effect of dietary supplementation with   Polyunsaturated fatty acids (different two ratio) on skin wound healing (in rat model). 

It is a scientifically interesting work,  well organized and described by the authors . Their experimental setup included various  methods:  digital planimetry ( for wound closure), blood sample analyses ( hematological and biochemical analyses) and tissue samples analyses  (fluorescence microscopy and electron microscopy). The results were discussed and suggested  that after a short-term (two-week) administration of fat emulsions  (vegetable sources!) containing higher ratio of n-3:n-6 PUFA, the oxidative stress parameters during wound healing could be affected.

The following minor revisions of this manuscript need take into consideration :

1. Introduction: line 37-38 ( Wound healing always proceeds in following phases: inflammation, proliferation and remodelling) completed as: Wound healing always proceeds in following phases: inflammation, proliferation (granulation) and maturation (remodelling).

2. The animals (Wistar male rats) included in the study were divided into three groups (E, F, C). I don't understand why the authors called the test groups "E" and "F" (???). I recommend renaming them (A and B), group C remaining the control group.

Author Response

Dear Reviewer,

thank you very much for your careful review and helpful comments. For point-to-point response to your comments, please, see the attachment.

Reviewer 3 Report

I suggest to put more emphasis on the obtained results in the discussione section. Statistical analysis shoudl be better defined.

Author Response

(The authors gave the same response as above.)

Reviewer 4 Report

The English must be thoroughly improved
for example, the term "cell detritus" should be replaced by "cell debris"  

Please provide more information on PUFAs in the introduction

How many ml of lipid emulsion did the rats receive ?

Was blood drawn from the tail vein?

Please provide a diagram showing the location of tissue sections taken for microscopic examination within the wound

There are no grids in Figure 2, so please remove the sentence: "Bars were 190
calibrated in cm"

Figures 3 and 4 - The microphotographs are blurred and out of focus, and the cell nuclei cannot be seen accurately. I propose to enlarge it and  place additional microphotographs showing the cells marked with arrows at a higher magnification

Fotomikrografy wykonane za pomocą mikroskopu elektronowego muszą być powiększone, ponieważ szczegóły obrazu nie są wyraźnie widoczne.

All Figures - Bars and their numerical value are very weakly visible

Author Response

Dear Reviewer,

thank you very much for your careful review and helpful comments which substantially improved the quality of our manuscript. For point-to-point response to your comments, please, see the attachment.

Round 2

Reviewer 1 Report

The authors have adequately responded to the comments of the reviewer.

Reviewer 4 Report

The authors have revised the manuscript according to my comments. 

This manuscript is a resubmission of an earlier submission. The following is a list of the peer review reports and author responses from that submission.